# MG-DIFF: A novel molecular graph diffusion model for molecular generation and optimization

Xiaochen Zhang[1]*, Shuangxi Wang[1], Ying Fang[1], Qiankun Zhang[2]

**1** School of Information Technology, Shangqiu Normal University, Shangqiu, Henan, People's Republic of China, **2** Neurology Department, Shangqiu Municipal Hospital, Shangqiu, Henan, People's Republic of China

* zhangxiaochen@nudt.edu.cn

## Abstract

Recent advancements in denoising diffusion models have revolutionized image, text, and video generation. Inspired by these achievements, researchers have extended denoising diffusion models to the field of molecule generation. However, existing molecular generation diffusion models are not fully optimized according to the distinct features of molecules, leading to suboptimal performance and challenges in conditional molecular optimization. In this paper, we introduce the MG-DIFF model, a novel approach tailored for molecular generation and optimization. Compared to previous methods, MG-DIFF incorporates three key improvements. Firstly, we propose a mask and replace discrete diffusion strategy, specifically designed to accommodate the complex patterns of molecular structures, thereby enhancing the quality of molecular generation. Secondly, we introduce a graph transformer model with random node initialization, which can overcome the expressiveness limitations of regular graph neural networks defined by the first-order Weisfeiler-Lehman test. Lastly, we present a graph padding strategy that enables our method to not only do conditional generation but also optimize molecules by adding certain atomic groups. In several molecular generation benchmarks, the proposed MG-DIFF model achieves state-of-the-art performance and demonstrates great potential molecular optimization.

## Introduction

Designing new molecules with desired properties has posed a longstanding challenge [1]. The main difficulty arises from the vastness of the synthetically accessible, drug-like chemical space, estimated to range between $10^{23}$ and $10^{60}$, making the exploration of molecules within this scope practically infeasible [2]. Traditional approaches rely on high-throughput screening (HTS) technologies to sift through a small fraction of the synthetically chemical space, typically containing millions of compounds [3]. Despite already being highly time-consuming and labor-intensive, these methods are still prone to failure due to the huge disparity between accessible and

**Data availability statement:** The MOSES dataset can be accessed at https://github.com/molecularsets/moses. The ZINC-250K dataset can be accessed at https://www.kaggle.com/datasets/basu369victor/zinc250k. Codes for MG-Diff can be accessed at https://github.com/zhang-xuan1314/MG-Diff.

**Funding:** This work was supported by the Key scientific research projects in higher education institutions of Henan province (24A520036), and the Henan Province Science and Technology Tackling Key Problems Project (242102111186). The study was approved by the university's review board. The funders had no role in study design, data collection and analysis, decision to publish, or preparation of the manuscript.

**Competing interests:** The authors have declared that no competing interests exist.

potential molecules [4]. To address this issue, machine learning approaches have emerged as a promising approach to efficiently explore chemical space, and identify novel drug candidates that have not been previously synthesized, offering a complementary strategy to traditional drug discovery [5].

Among various types of machine learning approaches, deep generative networks offer a promising avenue for the automated design of molecules with desired properties [6]. Deep generative networks have made significant strides in general data domains such as computer vision and natural language processing [7]. Such methods have also been employed to learn the distribution of molecules from a large set of molecules and therefore are capable of generating entirely new molecules through sampling the learned distribution [8,9]. This type of model does not require complex predefined rules and can automatically explore the molecular data space to find new candidate molecules. It can also perform conditional molecular generation and molecular optimization tasks and thus has attracted significant attention [10].

Currently, molecular generation models primarily rely on sequence modeling techniques, utilizing autoregressive approaches to gradually construct molecules. A prevalent approach involves leveraging language models or text-based VAE models to iteratively generate molecular string representations, such as SMILES [8,11] or SELFIES [12]. Alternatively, sequential models can also be used to generate molecular graphs by progressively adding atoms, bonds, or molecular fragments [13–15]. In addition to sequence modeling, there is a growing interest in non-autoregressive graph generation techniques that employ VAEs, GANs, or normalizing flows [16–18]. While promising, these non-autoregressive approaches have yet to rival the performance levels demonstrated by autoregressive models, which excel at modeling the distributions of complex data [16].

The emergence of diffusion models has significantly propelled non-autoregressive generation models in various domains, including text, images, audio, and point clouds [19,20]. Inspired by this progress, Niu et al. utilized continuous value thresholding to generate adjacency matrices, indicating edges [21]. Haefeli et al. first demonstrated the effectiveness of discrete diffusion for graph generation and developed a model specifically tailored for unattributed graphs [22]. Concurrently, the DiGress model introduced a 2D molecular generation model based on discrete denoising diffusion, achieving notable advancements compared to preceding models [23]. In the meantime, many works have introduced denoising diffusion models for the generation of molecules in 3D [24]. These models generate atomic positions instead of graph structures, representing a promising alternative. However, this approach necessitates the utilization of conformer data for training.

This article focuses on the utilization of diffusion models for the direct generation of 2D molecular structures. Presently, the DiGress model stands as the state-of-the-art diffusion model for directly generating molecular structures. While DiGress has demonstrated the effectiveness of discrete diffusion for 2D molecular graphs, it suffers from two major limitations. First, its uniform discrete noise scheduler is aggressive and may introduce uninformative or conflicting perturbations, making the reverse estimation unstable. Second, DiGress relies on fixed-size sampling of molecular

graphs, which becomes problematic for conditional generation or optimization, where target molecule size is typically unknown.

To overcome these issues, we propose MG-DIFF with a mask-and-replace corruption strategy and graph padding mechanism, enabling more stable generation and flexible control over molecule size and structure. Firstly, we introduce a novel discrete diffusion scheduler with a masking and replacement strategy specifically tailored for molecular structures. During the training stage, we propose to gradually mask atoms and bonds in molecules. Meanwhile, each atom and bond also has a small probability of changing to other types. In this way, During the denoising phase, the network can focus primarily on predicting the masked tokens while also allocating limited resources to modifying the unfit atoms or bonds. Compared to the conventional replace-only diffusion strategy, the masked tokens effectively direct the network's attention to the masked areas and thus greatly reduce the number of atom and bond combinations to be examined. This mask-and-replace diffusion strategy can significantly accelerate the convergence of the network. Secondly, we utilize a graph transformer model with a random node initialization strategy. A graph transformer is a type of graph neural network that utilizes transformer architectures and global attention mechanisms [25]. Graph Neural Networks (GNNs) are inherently constrained in their expressive capacity, as they are typically no more powerful than the first-order Weisfeiler-Lehman test (1-WL) [26]. To overcome this fundamental limitation, researchers have proposed augmenting GNNs with random node initialization, a technique involving the embedding of graph node features with random noise [27]. This augmentation strategy allows GNNs to preserve the permutation-invariance characteristic of regular GNNs while introducing a strong inductive bias. This bias facilitates the effective learning and integration of both local and global graph features. Additionally, RNI has been shown to effectively alleviate the issue of over-smoothing, a critical problem that significantly impacts the performance of graph-based models [28]. Lastly, we present a graph padding strategy that not only enables conditional generation but also facilitates molecule optimization through targeted local modifications. We propose to introduce a certain number of padding nodes into molecular graphs to ensure that all molecular graphs have the same number of nodes. It is worth noting that these padding nodes are not connected to any other nodes. During generation, we simply remove the padding nodes and identify the largest connected subgraph as the generated result. The advantage of this approach is that the model automatically captures the distribution of node numbers in molecular graphs, rather than sampling from the training data. This is highly beneficial for molecular conditional generation and optimization tasks, as the number of nodes in the target molecule cannot be predetermined. Experimental findings obtained from a diverse range of molecular generation benchmarks conclusively demonstrate the superior performance of the MG-DIFF model when compared to previous methodologies, firmly establishing it as the forefront of current research. Moreover, its promising potential for molecular optimization highlights intriguing prospects for further exploration and advancement in future research endeavors.

## Methods

### Background: Diffusion models

to better demonstrate our stragies, we first give a brief introduction to denoising diffusion model. The diffusion models are latent variable generative models, distinguished by their forward and reverse Markov processes. The forward process corrupts the original data $x_0 \sim q(x_0)$ into a sequence of progressively noisy latent variables $x_{1:T} = x_1, x_1, ..., x_T$ through a predetermined transition distribution $q(x_{1:T}|x_0) = \prod_{t=1}^{T} q(x_t|x_{t-1})$. The noisy schedule will be delicately designed to make $x_T$ become a certain pure noise distribution. On the flip side, the acquired reverse Markov process $p_\theta(x_{0:T}) = p(x_T) \prod_{t=1}^{T} p_\theta(x_{t-1}|x_t)$ begins with the prior noise distribution and progressively denoises the latent variables, leading toward the original data distribution. In this paper, we focus on discrete diffusion model.

### Discrete diffusion model

In the following content, we will mainly focus on the discrete denoising diffusion model. For scalar discrete random variables with $K$ categories $x_t, x_{t-1} \in \{1, ..., K\}$, the forward transition probabilities can be described using

matrices: $[\mathbf{Q}_t]_{ij} = q(x_{t-1} = j | x_t = i)$. Then the forward Markov diffusion process for the whole token sequence can be written as follows:

$$q(x_t|x_{t-1}) = \mathbf{v}^T(x_t)\mathbf{Q}_t\mathbf{v}(x_{t-1}) \tag{1}$$

where $\mathbf{v}(x)$ is a one-hot column vector whose length is K and only the entry $x$ is 1. The categorical distribution over $x_t$ is given by the vector $\mathbf{Q}_t\mathbf{v}(x_{t-1})$.

Importantly, leveraging the property of Markov chains [29], it is possible to marginalize out the intermediate steps and directly derive the probability of $x_t$ at any arbitrary timestep from $x_0$ as:

$$q(x_t|x_0) = \mathbf{v}^T(x_t)\overline{\mathbf{Q}}_t\mathbf{v}(x_0),\ \textit{with}\ \overline{\mathbf{Q}}_t = \mathbf{Q}_t\mathbf{Q}_{t-1}\cdots\mathbf{Q}_1. \tag{2}$$

To optimize the generative model $p_\theta(x_0)$ to closely match the underlying data distribution $q(x_0)$, a common approach is to maximize a variational upper bound on the negative log-likelihood:

$$L_{\text{vlb}} = E_{q(x_0)}\left[\underbrace{D_{KL}\left[q(x_t|x_0)||p(x_T)\right]}_{L_T} + \sum_{t=2}^{T}E_{q(x_t|x_0)}\underbrace{\left[D_{KL}\left[q(x_{t-1}|x_t,x_0)||p_\theta(x_{t-1}|x_t)\right]\right]}_{L_{t-1}}\right.$$
$$\left.-\underbrace{E_{q(x_t|x_0)}\left[\log p_\theta(x_0|x_1)\right]}_{L_0}\right] \tag{3}$$

where $q(x_{t-1}|x_t,x_0)$ is the posterior transition distribution which can be written in closed form as follows:

$$q(x_{t-1}|x_t, x_0) = \frac{q(x_t|x_{t-1},x_0)q(x_{t-1}|x_0)}{q(x_t|x_0)}$$
$$= \frac{(\mathbf{v}^T(x_t)\mathbf{Q}_t\mathbf{v}(x_{t-1}))(\mathbf{v}^T(x_{t-1})\overline{\mathbf{Q}}_{t-1}\mathbf{v}(x_0))}{\mathbf{v}^T(x_t)\overline{\mathbf{Q}}_t\mathbf{v}(x_0)} \tag{4}$$

In the loss formula, the $L_T$ term can be optimized by setting $p(x_T)$ as the the prior noise distribution and does not need training. In the left term, we need to train a denoising model $p_\theta(x_{t-1}|x_t)$ to estimate the posterior transition distribution. Early models attempted to directly predict $x_{t-1}$ from $x_t$. However, both $x_t$ and $x_{t-1}$ are sampled from the diffusion trajectory and thus exhibit significant variances. This poses challenges for models that attempt to directly predict $x_t$ based on $x_{t-1}$, leading to difficulties in training. Ho et al [30] proposed to use the noiseless target data $x_0$ as the target of the denoising network, effectively eliminating a significant source of label noise and demonstrating superior generation quality. In the setting, the network predicts the noiseless target distribution $p_\theta(\tilde{x}_0|x_t)$ at each reverse step and the reverse transition distribution can be computed as follows:

$$p_\theta(x_{t-1}|x_t) = \sum_{\tilde{x}_0=1}^{K} q(x_{t-1}|x_t,\tilde{x}_0)p_\theta(\tilde{x}_0|x_t) \tag{5}$$

Based on the above reparameterization trick, an auxiliary denoising objective can be introduced, which encourages the network to predict noiseless token $x_0$:

$$L_{\text{CE}} = -\log p_\theta(x_0|x_t) \tag{6}$$

In our experiments, we find this loss combined with the $L_{\text{vlb}}$ loss can greatly improve the generation quality.

## Mask-and-replace diffusion strategy

Prior research proposed to utilize uniform noise to corrupt the categorical distribution. However, employing uniform diffusion for data corruption can be quite aggressive, posing challenges for reverse estimation. Firstly, for molecular data, an atom or bond token may be replaced by an entirely uncorrelated category, resulting in a sudden semantic shift. Secondly, the network must exert additional effort to find replaced tokens before rectifying them. In fact, due to semantic conflicts within the local context, reverse estimation for different tokens may become competitive, making it difficult to identify reliable tokens.

To address the shortcomings of uniform diffusion, we draw inspiration from mask language modeling and propose to utilize the mask and replace diffusion strategy. This involves stochastically masking some tokens so that the reverse network can explicitly identify corrupted locations. Specifically, we introduce an additional special token, [MASK], so that each token now has $(K+1)$ discrete states. The mask diffusion is defined as follows: Each ordinary token has a probability of $\alpha_t$ being unchanged, a probability of $\gamma_t$ being replaced by the [MASK] token, and a chance of $\beta_t$ being uniformly replaced.

$$
Q_t = \begin{bmatrix}
\alpha_t + \beta_t & \beta_t & \beta_t & \cdots & 0 \\
\beta_t & \alpha_t + \beta_t & \beta_t & \cdots & 0 \\
\beta_t & \beta_t & \alpha_t + \beta_t & \cdots & 0 \\
\vdots & \vdots & \vdots & \cdots & 0 \\
\gamma_t & \gamma_t & \gamma_t & \cdots & 1
\end{bmatrix}
\tag{7}
$$

Of course, relying solely on the mask-only strategy is also problematic. This is because the model's sampling process may lead to incorrect predictions for masked parts. Mask-only diffusion models cannot modify the incorrectly predicted parts. Therefore, introducing a small amount of uniform noise is highly necessary, as it allows the model to make appropriate adjustments to the prediction errors. Naturally, the level of uniform noise is minimal, with the model primarily focusing on the recovery of mask tokens and then allocating a small number of capacity to correcting prediction errors. This approach is particularly suitable for molecules, where the model initially learns to recover the main structure of the molecule, and then dedicates a small amount of capacity to fixing incompatible parts.

Compared to uniform corruption strategies, the proposed mask-and-replace approach offers two key theoretical advantages. First, inspired by the success of masked language modeling in NLP (e.g., BERT), this strategy enables the model to explicitly learn the context-sensitive structure of molecules by focusing on recovering the corrupted positions. This significantly reduces the uncertainty in the reverse process by clearly indicating which tokens require prediction. Second, the masking strategy provides a more informative training signal than uniform replacement, which can often lead to abrupt semantic disruptions in molecular graphs due to the replacement of atoms or bonds with incompatible types. From an information-theoretic standpoint, this strategy increases the signal-to-noise ratio during learning and accelerates convergence by constraining the output space. We hypothesize that this alignment with domain semantics makes the mask-and-replace process more suitable for molecular graph generation than vanilla categorical noise.

## Molecular graph diffusion model(MG-DIFF)

In this section, We introduce how to use the diffusion model for molecular graph generation. Molecular graphs can be represented by categorical attributes of the constituent atoms and bonds. The key atomic attributes can be divided into atomic types and atomic charges. We use ai to denote one hot encoding form of the atom type and ci to denote one hot encoding form of the atom charge. These encodings can be organized into matrix forms as $\mathbf{A} \in R^{n \times d_A}$, $\mathbf{C} \in R^{n \times d_C}$ for one molecule, where n is the number of atoms and dA, dC are corresponding one hot encoding dimensions. Similarly, We use eij to denote one hot encoding form of the edge type and form it into matrix form $\mathbf{E} \in R^{n \times n \times d_E}$. In this way, a molecular graph can be represented as G=<A,C,E>

For bonds, we introduce a "no bond" type to indicate the absence of a bond between two atoms. Additionally, we utilize the trick of **atomic padding,** where a certain number of atoms are padded to molecules, enabling all molecules to have the same graph size. The advantage of this approach is that the model can automatically capture the distribution of the number of atoms in molecules. To achieve this, we include a "<PADDING>" category within the atomic types. Padding atoms do not have bonds with any other atoms. Besides, a "<MASK>" category is included in atom types, atom charges, and bond types for the mask and replace noise strategy. While the padding strategy has a relatively limited impact on unconditional molecular generation performance (e.g., validity and novelty metrics remain largely unchanged when padding is removed), it is critical for enabling molecular optimization. Without a fixed-size representation across samples, our model would not be able to introduce additional atoms into the molecular graph in a principled manner. The padding nodes serve as placeholder positions that can be selectively activated during reverse diffusion, thus allowing for the addition of atomic groups required for property-driven molecule refinement.

In the forward process, we employ the mask and replace noise approach and design three transition matrices that simultaneously introduce noise into the atomic types, atomic charges, and bonds. These transition matrices are meticulously crafted to ensure a consistent level of noise across these three variables within each time step. Subsequently, adding noise to form $G_t$ is achieved simply by sampling atoms, charges, and bond types from the categorical distribution defined by the respective probabilities.

$$q(G_t|G_{t-1}) = (\mathbf{A}_{t-1}\mathbf{Q}_t^A, \mathbf{C}_{t-1}\mathbf{Q}_t^C, \mathbf{E}_{t-1}\mathbf{Q}_t^E) \; and \; q(G_t|G) = (\mathbf{A}_{t-1}\overline{\mathbf{Q}}_t^A, \mathbf{C}_{t-1}\overline{\mathbf{Q}}_t^C, \mathbf{E}_{t-1}\overline{\mathbf{Q}}_t^E) \tag{8}$$

Because the molecular graphs are undirected, we just apply noise to the lower-triangular part of **E** and then symmetrize it.

In the training process, we employ a graph transformer model to predict the clean graph based on the noisy graph. Detailed architecture information about this model is shown in the S1 text. To train the model, we utilize the variational lower bounds of the three variables, along with auxiliary losses.

$$L = w_1 L_{vb}^A + w_2 L_{vb}^C + w_3 L_{vb}^E + w_4 L_{CE}^A + w_5 L_{CE}^C + w_6 L_{CE}^E \tag{9}$$

where $w_1, w_2, ..., w_6$ is the relative weights scalar to balance each loss.

Once the network has been trained, it can be leveraged to generate new molecular graphs. To accomplish this, it is necessary to estimate the reverse diffusion transition $p(G_{t-1}|G_t)$. We can utilize equation 5 separately for atom types, atom charges, and bond types to achieve this objective. Then we can start a fixed-size graph in which atom type, charge type, and bond type all belong to the "<mask>" category and gradually use the reverse diffusion transition to generate a molecule.

## Conditional generation

To facilitate conditional diffusion within our diffusion model, we train the denoising network $p(G|G_t,c)$ conditioned on the target property of value $c$. The training and sampling process resembles that of the unconditional model, with the only distinction being that the conditional model takes the property value as input and concatenates it with the denoising model's node features. Our molecule padding strategy is quite useful in conditional generation because the atomic distribution of molecules is often unknown under specific conditions, and our model can autonomously learn from the data.

## Molecular optimization

Our diffusion model provides a natural way for molecular optimization. For a certain property, we first train a conditional generation model $p(G|G_t,c)$ and sample from this model based on a specified property value. In the sample stage, given a start molecular $S=<\mathbf{A}_s, \mathbf{C}_s, \mathbf{E}_s>$ with $n_s$ atoms, we can make sure S showed in the generated graph by masking the

generated node and edge feature tensor at each reverse iteration step. As our model is permutation equivariant, it does not matter what entries are masked: we therefore choose the first $n_s$ ones. After sampling $G_{t-1}$, we update **A**, **C,** and **E** using:

$$\mathbf{A}^{t-1} = \mathbf{M}_A \odot \mathbf{A}_S + (1 - \mathbf{M}_A) \odot \mathbf{A}^{t-1}, \mathbf{C}^{t-1} = \mathbf{M}_C \odot \mathbf{C}_S + (1 - \mathbf{M}_C) \odot \mathbf{C}^{t-1},$$
$$\mathbf{E}^{t-1} = \mathbf{M}_E \odot \mathbf{E}_S + (1 - \mathbf{M}_E) \odot \mathbf{E}^{t-1}$$

$$(10)$$

where $\mathbf{M}_A \in R^{n \times d_A}$, $\mathbf{M}_C \in R^{n \times d_C}$ and $\mathbf{M}_E \in R^{n \times n \times d_E}$ are masks indicating the $n_s$ first nodes.

## Training details

We trained all models using the AdamW optimizer with a learning rate of 1e-4 and weight decay of 1e-4. The batch size was set to 128. The MG-DIFF model utilizes a Graph Transformer architecture with 6 layers, a hidden size of 256, and 8 attention heads. Dropout with a rate of 0.1 was applied to prevent overfitting. All weights were initialized using the Xavier initialization strategy.

The model was trained for 500 epochs with a batch size of 128. the maximum norm of the gradient is clipped to 1 to keep the stability of training. The training was conducted on a single NVIDIA 4090 GPU with 24GB memory. The models were selected based on the best validation loss checkpoint. No pretraining was used; all models were trained from scratch.

## Results

### Unconditional molecular generation

**Data.** In this research, we directly utilized the MOSES [31] and ZINC-250K [32] datasets, which are well-curated benchmark datasets. Each dataset was randomly split into the training, validation, and test datasets with a ratio of 8:1:1. The molecular properties including QED, SAS, LogP, and TSPA are calculated by the open-source software RDKit [33].

**Baselines.** We compared our method to various approaches: (1)LatentGAN [34], a method that pre-trains an autoencoder to map SMILES structures onto latent vectors and further train a generative adversarial network to capture the distribution of the latent vectors; (2) JT-VAE [15], a method uses the junction tree variational autoencoder to generates molecular graphs; (3) VAE [35], a method that utilizes an encoder and a decoder to infer a mapping from the high-dimensional molecular data representation onto a lower-dimensional space and back; (4) MolGPT [8], a method that utilizes the transformer-decoder on the next token prediction task to capture the distribution of molecular SMILES. (5) DiGress [23], a method that uses the discrete denoising diffusion method for molecular graph generation.

**Metrics.** We utilized the following widely used metrics to evaluate the methods: Validity, Novelty, Diversity, Internal Diversity(IntDiv$_p$), Quantitative Estimation of Drug-Likeness (QED), and Synthetic Accessibility Score (SAS). Detailed definitions of these metrics can be found in the S1 Text.

For each method, we generated 10K molecules to calculate the metrics. All metrics except validity are computed on the set of valid molecules generated by the model. The model performance is reported in Table 1 and Table 2.

Higher values of validity, uniqueness, and novelty mean that the generation model can learn the molecular composition rules without overfitting the training data simultaneously. Based on Table 1 and Table 2, our model demonstrates a more comprehensive performance. Baseline models often excel in only one or two of these three metrics. For instance, Latent-GAN, Digress, and VAE perform poorly in terms of validity, while JT-VAE and MolGPT exhibit inferior novelty. Our model approaches near-optimal performance across all three metrics, without any noticeable shortcomings. This strongly indicates that our model can effectively capture the structural composition rules of molecules to generate valid molecules and can adequately explore molecular space.

 

**Table 1. The generation performance comparison on the ZINC-250K dataset.**

| Model | Validity(↑) | Unique (↑) | Novel (↑) | IntDiv1(↑) | IntDiv2(↑) | SAS (↓) | QED (↑) |
|---|---|---|---|---|---|---|---|
| LatentGAN | 0.863 | 0.951 | 0.969 | 0.801 | 0.792 | 3.124±0.562 | 0.709±0.074 |
| JT-VAE | 1.0 | 1.0 | 0.893 | 0.863 | 0.852 | 3.095±0.594 | 0.721±0.081 |
| VAE | 0.870 | 0.998 | 0.991 | 0.856 | 0.849 | 3.243±0.557 | 0.711±0.089 |
| MolGPT | 0.973 | 0.981 | 0.864 | 0.851 | 0.843 | 3.471±0.634 | 0.704±0.094 |
| DiGress | 0.852 | 1.0 | 0.999 | 0.855 | 0.849 | 3.379±0.627 | 0.729±0.093 |
| MG-Diff | 0.964 | 0.994 | 0.985 | 0.891 | 0.881 | 3.825±1.143 | 0.700±0.127 |

**Table 2. The generation performance comparison on the MOSES dataset.**

| Model | Validity(↑) | Unique(↑) | Novel(↑) | IntDiv1(↑) | IntDiv2(↑) | SAS(↓) | QED(↑) |
|---|---|---|---|---|---|---|---|
| LatentGAN | 0.897 | 0.997 | 0.949 | 0.857 | 0.851 | 2.831±0.561 | 0.803±0.107 |
| JT-VAE | 1.0 | 0.999 | 0.914 | 0.855 | 0.849 | 2.753±0.586 | 0.801±0.102 |
| VAE | 0.977 | 0.998 | 0.695 | 0.856 | 0.849 | 2.636±0.459 | 0.806±0.094 |
| MolGPT | 0.994 | 1.0 | 0.897 | 0.857 | 0.851 | 2.884±0.582 | 0.750±0.113 |
| DiGress | 0.857 | 1.0 | 0.950 | 0.845 | 0.839 | 2.693±0.551 | 0.802±0.092 |
| MG-Diff | 0.977 | 0.999 | 0.991 | 0.881 | 0.879 | 3.128±1.033 | 0.782±0.119 |

Internal diversity scores provide an understanding of the extent of chemical space traversed by different models. Our model achieves the best performance in terms of Internal diversity scores, which is over 2% higher than that of other models. This indicates that our model can more efficiently explore the molecular space.

SAS and QED measure the drugability and synthesizability of molecules generated by models. Our model performs slightly less well on these two metrics, but still within a reasonable range. While this might initially raise concerns regarding stability and consistency, we interpret this as a reflection of the model's capacity to generate structurally diverse molecules across a wide chemical space. In practical applications, such diversity is valuable for early-stage exploration, enabling the discovery of novel candidates. However, it may also result in less consistency in property quality across samples. This trade-off suggests that MG-DIFF may benefit from additional post-generation filtering or reinforcement-based fine-tuning strategies to improve property control in downstream drug design tasks. Additionally, future work could explore integrating property-aware loss functions to reduce variance while maintaining molecular diversity.

## Conditional generation

In biology and chemistry, various processes rely on molecules possessing specific property values to fulfill their designated functions. For instance, for molecules to traverse this barrier and interact with receptors in the central nervous system, a TPSA value below 90 Å$^2$ is usually required. So we further evaluate the capability of our proposed MG-Diff model in conditional generation. Specifically, we tested the conditional generation performance on four molecular properties in the ZINC-250K dataset: LogP, TPSA, SAS, and QED. LogP is the log of the partition coefficient of a solute between octanol and water. TPSA(topological polar surface area) of a molecule is defined as the surface sum over all polar atoms or molecules. SAS and QED as introduced before, measure the drugability and synthesizability of molecules generated by models. These four properties are the key factors determining the drug-likeness of molecules. Of course, our model can also be applied to other molecular properties as well. For each conditional test, we generate 10,000 molecules to evaluate conditional generation performance.

Fig 1 illustrates the distribution of the corresponding properties of conditionally generated molecules. Table 3 further provides key metrics such as validity, uniqueness, novelty, and the mean average deviation (MAD) for each property. As

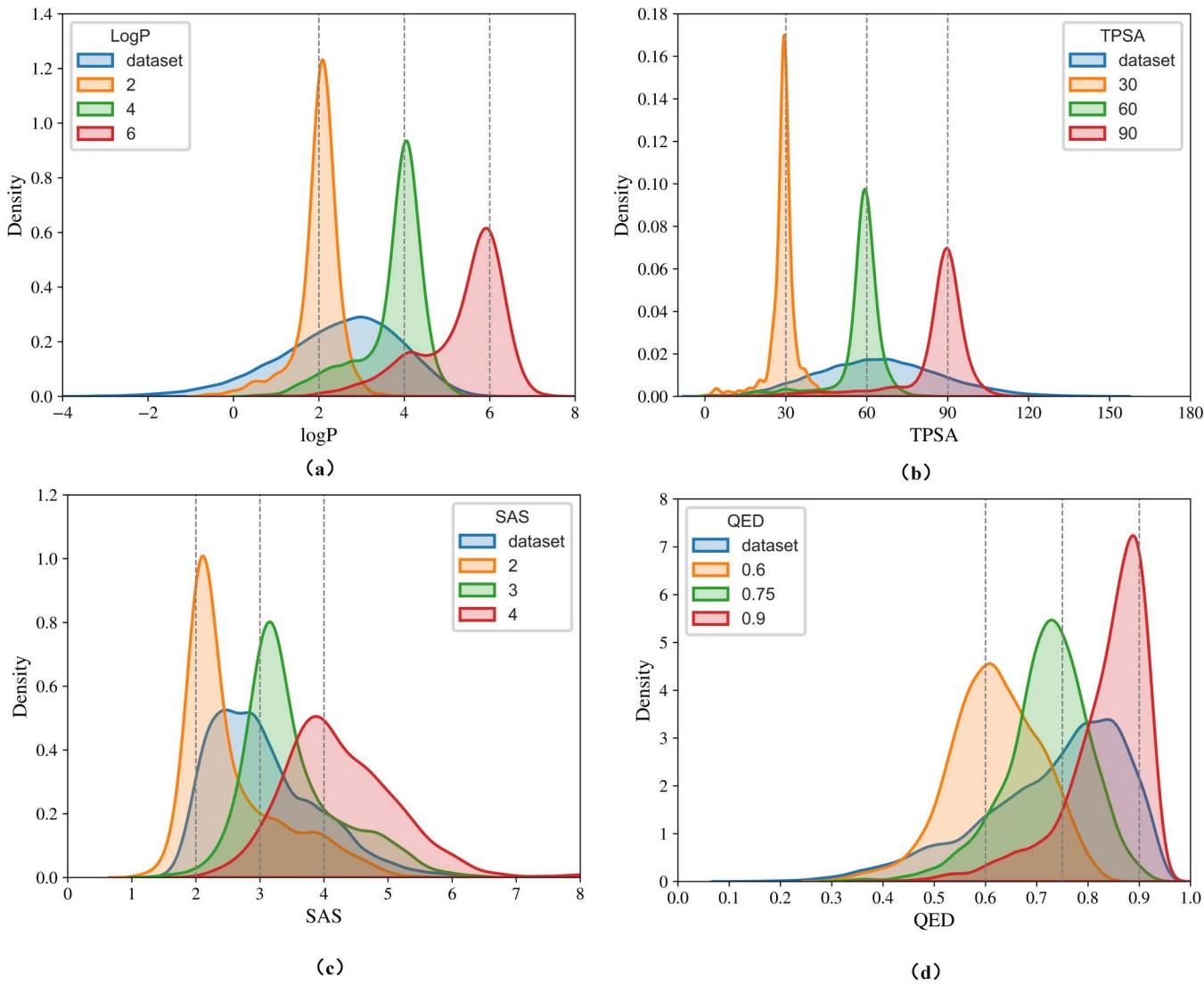

**Fig 1. Distribution of property of generated molecules conditioned on the (a) logP (b) TPSA (c) SAS (d) QED property.** The distribution depicted using a blue line corresponds to the original ZINC-250K dataset.

**Table 3. The conditional generation performance of MG-DIFF.**

| condition | validity | unique | novelty | MAD |
|---|---|---|---|---|
| logP | 0.935 | 0.994 | 0.993 | 0.574 |
| SAS | 0.943 | 0.991 | 0.989 | 0.724 |
| QED | 0.963 | 0.995 | 0.993 | 0.078 |
| TSPA | 0.932 | 0.990 | 0.991 | 5.436 |

can be seen in Fig 1, the property distributions are clustered around the desired values. As shown in Table 3, The validity, uniqueness, and novelty remain quite satisfactory. The stochastic nature of the mask-and-replace scheduler and the inherent difficulty of tightly controlling complex properties. We also note that while MAD indicates dispersion, the property

distributions still cluster near the target values, which supports the effectiveness of our method. Future improvements could include adaptive noise scheduling or property-aware loss functions.

## Molecular optimization

The proposed model can effectively optimize specific molecules by augmenting their pertinent properties through the addition of supplementary atomic groups. This enhancement is achieved by introducing masked atoms into the original molecule and recovering these masked atoms throughout the diffusion generation process. For a comprehensive understanding of the methodology, please refer to the Methods section. Notably, this optimization model does not require re-training the model and can utilize the previously trained conditional generation model from the preceding section. By setting the desired molecular property value, the diffusion model can add a diverse atomic group to achieve the targeted property.

Firstly, we focused on the LogP property. Fig 2 illustrates the outcomes of two molecular optimization processes. LogP serves as a crucial metric for quantifying the hydrophilicity or hydrophobicity of a molecule. Generally, molecules with higher polarity exhibit lower LogP values, indicating their affinity towards water. Conversely, molecules with lower polarity display higher LogP values, indicating a reduced tendency to interact with water. Our model can precisely modify the LogP property by incorporating polar or non-polar functional groups to achieve the predefined target.

Nevertheless, it's important to note that our model is limited to adding atoms to the original molecule but not removing them, which may constrain its ability to optimize certain properties. As shown in Fig 3, when optimizing molecules for TSPA or QED, we observed that while unlimited optimization is unattainable, significant enhancements can be made based on well-defined objectives. For instance, our model effectively reduces polarity by introducing atoms to polar functional groups during TSPA optimization. Similarly, in the case of QED, the model learns to integrate common atomic groups to modify the molecular druggability. However, our model encounters challenges in effectively optimizing the SAS property due to the intricate relationship between a drug's synthesizability and molecular complexity. Nevertheless, our current optimization approach is inherently limited to atom addition and cannot perform atom deletion. This restricts the model's ability to reduce molecular complexity or remove substructures that negatively impact properties such as synthetic accessibility (SAS) or toxicity. For example, when optimizing toward lower SAS scores, the inability to eliminate complex or unstable moieties may hinder further improvement. We believe that enabling atom or bond deletion would be a valuable enhancement for practical medicinal chemistry applications. One potential direction is to incorporate reversible masking or learned structural pruning mechanisms during the reverse diffusion process.

## Discussion

In this work, we introduced a novel discrete diffusion model called MG-DIFF for molecular generation and optimization. In contrast to autoregressive approaches and recent 3D-based generative models, MG-DIFF provides a scalable and interpretable framework for 2D molecular generation, with strong support for structure-aware conditional design and optimization. Leveraging a masked and replace strategy as the data corruption rule, this model proves instrumental in deciphering the intricate composition rules governing molecules. Furthermore, we innovate with the introduction of the atom padding strategy, which empowers the diffusion model to autonomously discern the distribution of atomic numbers within molecules. Compared to DiGress, which is the most closely related method to our work, MG-DIFF introduces three key improvements. The mask-and-replace corruption strategy avoids the harshness of uniform noise and enables targeted denoising, improving convergence and stability. The graph transformer with random node initialization enhances the model's expressiveness and helps capture long-range dependencies in molecular graphs. Finally, the graph padding mechanism allows MG-DIFF to handle variable-sized molecules more effectively, a crucial capability for conditional generation and optimization tasks. These improvements not only address the core limitations of DiGress but also broaden the applicability of diffusion models in practical molecular design scenarios. Through extensive benchmarking experiments, MG-DIFF exhibits remarkable performance metrics across various dimensions, showcasing exemplary

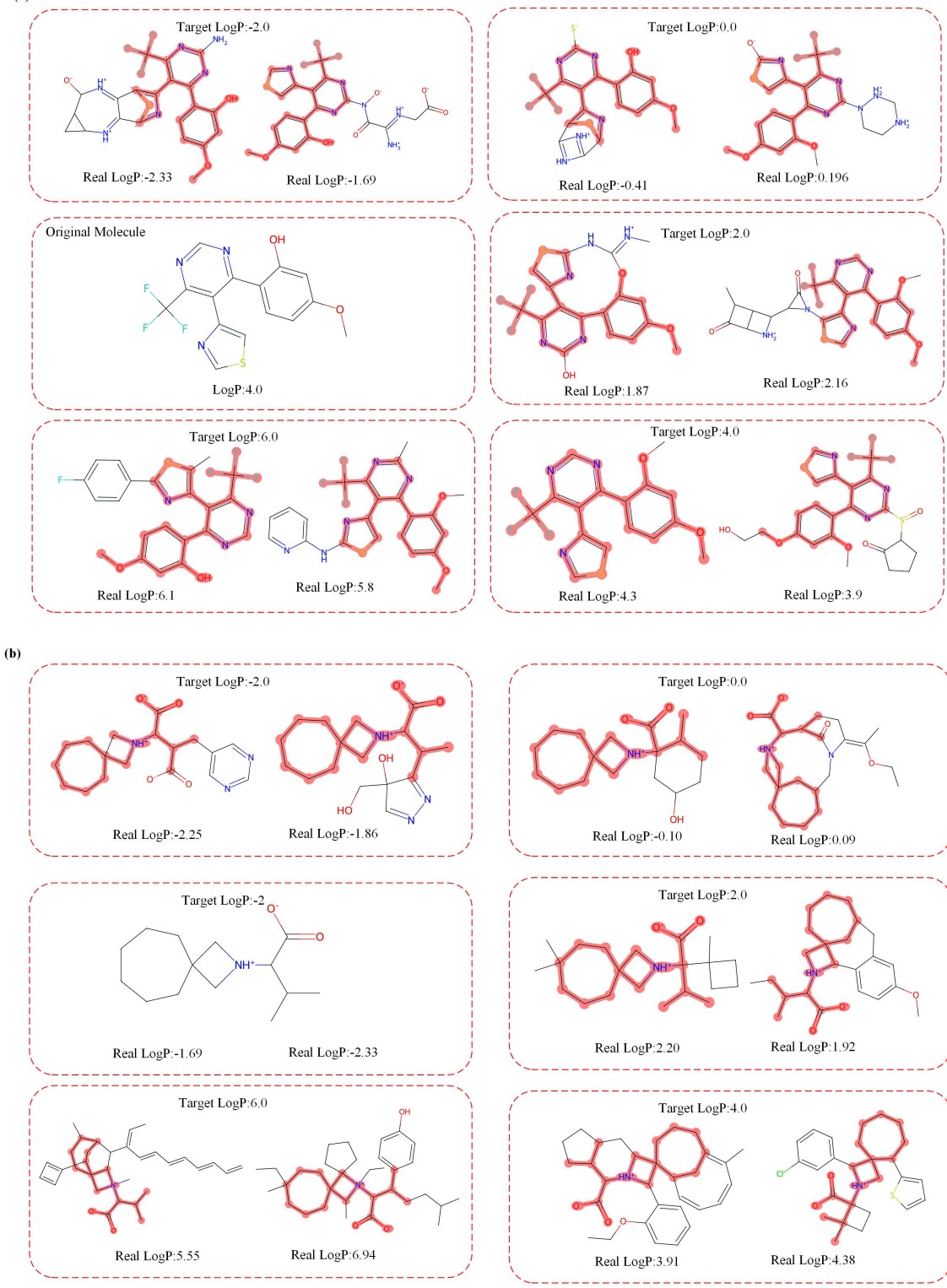

**Fig 2. Optimization of the LogP value based on two randomly picked molecules.** The original molecules are highlighted in the generated molecules.

(a) Optimizing TSPA with a target of 30.

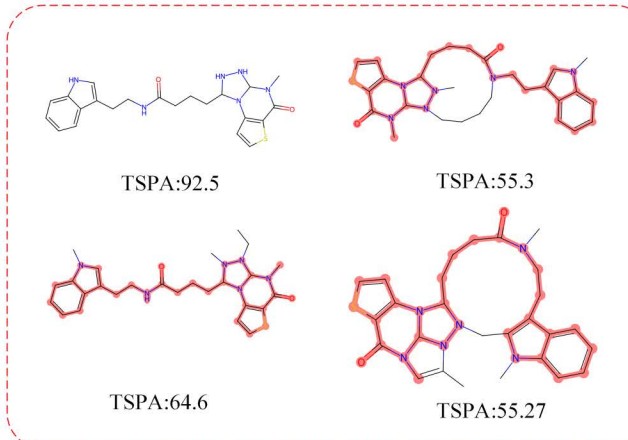

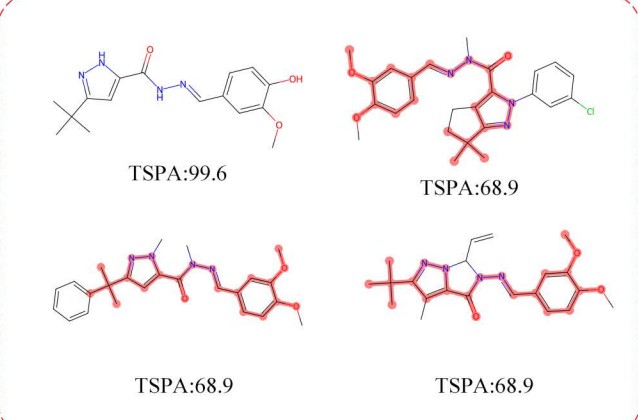

(b) Optimizing QED with a target of 0.9

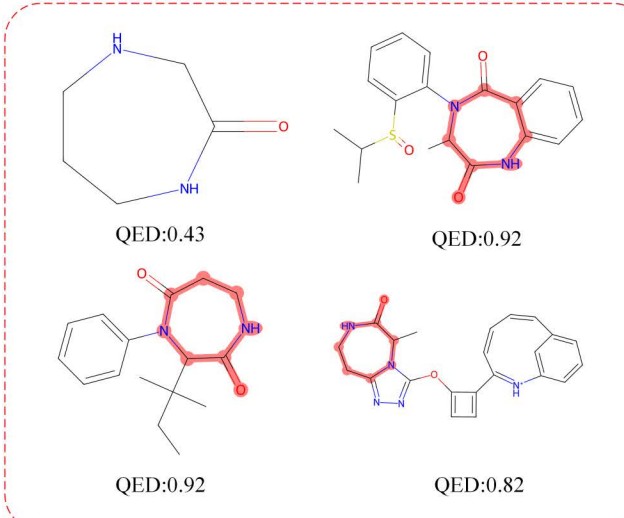

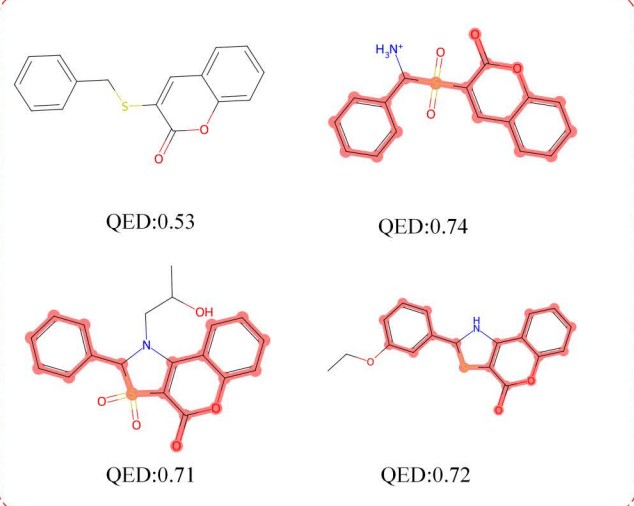

**Fig 3. Optimization of the (A) TSPA and (B) QED value based on two randomly picked molecules separately.** In each subgraph, the left-upper corner is the original molecule, and the original molecule is highlighted in the generated molecules.

validity, uniqueness, and novelty scores on benchmark datasets such as MOSES and ZINC-250K. while ZINC-250K and MOSES are standard benchmarks, MG-DIFF is designed to be general and modular. For domain-specific applications (e.g., ChEMBL or PubChem), fine-tuning or transfer learning would be necessary to adapt to new property distributions or chemical spaces. Notably, our model excels in IntDiv scores on both datasets, underscoring its robustness and efficacy. Moreover, our findings reveal MG-DIFF's proficiency in acquiring higher-level chemical representations, evident through its adeptness in molecular property control. A key limitation of MG-DIFF lies in its inability to delete atoms or substructures during molecular optimization. This constraint prevents the model from simplifying molecular graphs or removing undesired chemical groups, which are common strategies in medicinal chemistry. Although our padding-based framework supports atom addition flexibly, future work may explore graph editing operations that allow for both addition and deletion, possibly through structure-aware masking or energy-guided diffusion processes.

MG-DIFF demonstrates the capability to generate molecules with conditional property values. Furthermore, our model facilitates molecular property optimization while preserving specific molecular components. In light of these achievements, we posit that MG-DIFF emerges as a formidable architecture for advancing the frontiers of novel molecular generation and optimization. Its versatility and performance highlight its potential as a valuable tool in molecular design and discovery.

## Supporting information

**S1 Text. Supplementary model configuration and metrics.** Includes detailed descriptions of the MG-DIFF model architecture, training hyperparameters, and definitions of evaluation metrics used in molecular generation and optimization tasks.
(DOCX)

## Acknowledgments

We acknowledge Yifan Dong for his carefully formatting advices.

## Author contributions

**Conceptualization:** XiaoChen Zhang, Ying Fang.

**Data curation:** Shuangxi Wang.

**Formal analysis:** XiaoChen Zhang, Shuangxi Wang, Qiankun Zhang.

**Software:** XiaoChen Zhang, Ying Fang.

**Supervision:** Qiankun Zhang.

**Validation:** Ying Fang.

**Visualization:** Shuangxi Wang, Qiankun Zhang.

**Writing – original draft:** XiaoChen Zhang.

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
