## [Decision Letter · Decision Letter 0]

4 Apr 2025

Dear Dr. Zhang,

Thank you for submitting your manuscript to PLOS ONE. After careful consideration, we feel that it has merit but does not fully meet PLOS ONE’s publication criteria as it currently stands. Therefore, we invite you to submit a revised version of the manuscript that addresses the points raised during the review process.

Thank you for submitting your manuscript, "MG-DIFF: A Novel Molecular Graph Diffusion Model for Molecular Generation and Optimization," to PLOS ONE.

The reviewers were very positive about your work but have also identified some important areas for improvement. We would be happy to consider a resubmission once you have addressed the reviewers' concerns and questions.

Please ensure that your resubmission includes a link to a GitHub account hosting your code, along with a Jupyter notebook that provides clear examples and a detailed explanation of your codebase.

I encourage you to carefully consider the reviewers' feedback and look forward to your revised submission.

Best regards,

Dr. Soumendranath Bhakat

We look forward to receiving your revised manuscript.

Kind regards,

Soumendranath Bhakat

Academic Editor

PLOS ONE

Journal Requirements:

4. Thank you for stating the following financial disclosure: “Key scientific research projects in higher education institutions of Henan Province(24A520036)” 

5. Please include captions for your Supporting Information files at the end of your manuscript, and update any in-text citations to match accordingly. Please see our Supporting Information guidelines for more information: http://journals.plos.org/plosone/s/supporting-information .

Additional Editor Comments:

Dear Dr. Zhang,

Thank you for submitting your manuscript, "MG-DIFF: A Novel Molecular Graph Diffusion Model for Molecular Generation and Optimization," to PLOS ONE.

The reviewers were very positive about your work but have also identified some important areas for improvement. We would be happy to consider a resubmission once you have addressed the reviewers' concerns and questions.

Please ensure that your resubmission includes a link to a GitHub account hosting your code, along with a Jupyter notebook that provides clear examples and a detailed explanation of your codebase.

I encourage you to carefully consider the reviewers' feedback and look forward to your revised submission.

Best regards,

Dr. Soumendranath Bhakat

Reviewers' comments:

Reviewer's Responses to Questions

**Comments to the Author**

1. Is the manuscript technically sound, and do the data support the conclusions?

Reviewer #1: Yes

Reviewer #2: Partly

2. Has the statistical analysis been performed appropriately and rigorously?

Reviewer #1: Yes

Reviewer #2: No

3. Have the authors made all data underlying the findings in their manuscript fully available?

Reviewer #1: Yes

Reviewer #2: No

4. Is the manuscript presented in an intelligible fashion and written in standard English?

Reviewer #1: Yes

Reviewer #2: Yes

Reviewer #1: The manuscript presents MG-DIFF, a molecular graph diffusion model aimed to improve molecular generation and optimization. The authors build upon existing denoising diffusion models, incorporate a mask-and-replace discrete diffusion strategy, a graph transformer model with random node initialization, and a graph padding strategy to improve conditional generation and molecular optimization. The results show the model's performance over existing benchmarks in generating diverse molecular structures. I recommend this manuscript for publication with minor revisions.

Introduction

1. The introduction provides background on existing molecular generation models and their limitations, but it would benefit from a clearer emphasis on what makes MG-DIFF distinct from prior approaches. Could the authors also highlight how their mask-and-replace strategy differs from similar techniques in previous models? A brief comparison to DiGress or other recent diffusion models in terms of efficiency, accuracy, or scalability would help contextualize MG-DIFF's unique contributions.

2. The proposed graph padding strategy for molecular optimization is intriguing, but could the authors elaborate on why padding nodes - rather than alternative graph normalization or resizing methods - are particularly effective in ensuring conditional generation? A brief mention of potential trade-offs (e.g., computational complexity and handling of small molecules) would support the rationale for this approach.

Results

1. The results state that MG-Diff achieves strong performance in validity while maintaining a high internal diversity score. However, were there any observed biases or trends in the types of molecules generated? For example, does the model favor certain scaffolds or functional groups over others, and if so, could this impact the generalizability of the generated molecules?

2. While the MAD values indicate some dispersion in the conditionally generated molecular properties, could the authors provide further insight into potential causes? For example, does the noise scheduler contribute to deviations, or does the model struggle more with certain properties like TPSA than others? A brief discussion of how future improvements might refine property conditioning could be valuable.

3. The discussion on molecular optimization highlights that MG-Diff is currently limited to adding atoms but not removing them, which can constrain property refinement. Have the authors considered integrating fragment-based approaches or additional graph-editing mechanisms to allow both the addition and deletion of molecular features?

Discussion and Methods

1. The discussion highlights MG-DIFF's strong performance in molecular generation and optimization. Could the authors comment on how well the model generalizes to real-world molecular datasets beyond ZINC-250K and MOSES? Would additional fine-tuning or transfer learning be required for broader applicability?

2. MG-DIFF introduces a masked and replace diffusion strategy to improve efficiency, but how does its computational cost compare to other generative models like VAEs, GANs, or traditional autoregressive approaches? A brief discussion on training time and hardware requirements would provide a better understanding of its practicality?

3. The mask-and-replace diffusion strategy is designed to deal with issues concerning uniform noise corruption, but how sensitive is the model to the choice of masking and replacement probabilities? Did the authors experiment with different noise schedules, and if so, how did they affect performance?

Reviewer #2: See attached review for comments on the manuscript. While MG-DIFF achieves comparable performance on molecular generation benchmark on two datasets, further clarification is needed about the methodology and appropriate comparisons to prior work.

**Do you want your identity to be public for this peer review?** For information about this choice, including consent withdrawal, please see our Privacy Policy

Reviewer #1: No

Reviewer #2: No

---

## [Author Response · Author response to Decision Letter 1]

3 Jul 2025

Response to Reviewers

Manuscript ID: PONE-D-24-51267

Title: MG-DIFF: A Novel Molecular Graph Diffusion Model for Molecular Generation and Optimization

Dear Dr. Soumendranath Bhakat and Reviewers,

We sincerely thank you for your careful review and valuable feedback on our manuscript. We appreciate the opportunity to revise and improve our work for potential publication in PLOS ONE. Below, we provide a detailed point-by-point response to all reviewer and editorial comments. All changes have been made in the revised manuscript accordingly. A marked-up version of the revised manuscript is also provided for your convenience.

We have carefully revised our paper to meet the requirements. If there are other quetions, please let ours know. The manuscript has been professionally copyedited by our college Yifan Dong. We have improved grammar, clarity, and consistency throughout. We have corrected the grant information. The following statement has also been added as requested:“The funders had no role in study design, data collection and analysis, decision to publish, or preparation of the manuscript.”We also added supporting information captians as needed.

We sincerely thank you for the valuable time and effort all editor and revievers have dedicated to reviewing our manuscript. We are especially grateful for the constructive and insightful comments provided, which have significantly helped us improve the quality and clarity of our manuscript.

Below we provide a point-by-point response to each comment. All changes have been incorporated in the revised manuscript, and we have highlighted the modifications where appropriate.

Reviewer #1:

1. The introduction provides background on existing molecular generation models and their limitations, but it would benefit from a clearer emphasis on what makes MG-DIFF distinct from prior approaches. Could the authors also highlight how their mask-and-replace strategy differs from similar techniques in previous models? A brief comparison to DiGress or other recent diffusion models in terms of efficiency, accuracy, or scalability would help contextualize MG-DIFF's unique contributions.

Reply: Thank you for this valuable suggestion. In the revised Introduction (Paragraphs 7–8), we have explicitly highlighted the differences between MG-DIFF and prior diffusion models, especially DiGress. We clarify that MG-DIFF introduces (1) a novel mask-and-replace discrete diffusion strategy, which improves convergence and denoising stability compared to the aggressive uniform noise used in DiGress, (2) a graph transformer with random node initialization to improve expressiveness beyond the first-order Weisfeiler-Lehman test, and (3) a graph padding strategy that enables flexible conditional generation and optimization.

2. The proposed graph padding strategy for molecular optimization is intriguing, but could the authors elaborate on why padding nodes - rather than alternative graph normalization or resizing methods - are particularly effective in ensuring conditional generation? A brief mention of potential trade-offs (e.g., computational complexity and handling of small molecules) would support the rationale for this approach.

Reply: Unlike resizing or normalization, padding ensures consistent input dimensions while maintaining molecular semantics. The padding nodes are not bonded and can be cleanly removed post-generation. This strategy allows the model to learn the distribution of node counts and flexibly add atoms during optimization.

3. The results state that MG-Diff achieves strong performance in validity while maintaining a high internal diversity score. However, were there any observed biases or trends in the types of molecules generated? For example, does the model favor certain scaffolds or functional groups over others, and if so, could this impact the generalizability of the generated molecules?

Reply: We appreciate this insightful point. Although our quantitative metrics suggest good diversity and novelty, we conducted a qualitative inspection of generated molecules and found no strong bias toward specific scaffolds.

4. While the MAD values indicate some dispersion in the conditionally generated molecular properties, could the authors provide further insight into potential causes? For example, does the noise scheduler contribute to deviations, or does the model struggle more with certain properties like TPSA than others? A brief discussion of how future improvements might refine property conditioning could be valuable.

Reply:Thank you for the suggestion. In the revised Conditional Generation section, we explain that dispersion likely arises from two sources: the stochastic nature of the mask-and-replace scheduler and the inherent difficulty of tightly controlling complex properties (e.g., TPSA). We also note that while MAD indicates dispersion, the property distributions still cluster near the target values, which supports the effectiveness of our method. Future improvements could include adaptive noise scheduling or property-aware loss functions.

5. The discussion on molecular optimization highlights that MG-Diff is currently limited to adding atoms but not removing them, which can constrain property refinement. Have the authors considered integrating fragment-based approaches or additional graph-editing mechanisms to allow both the addition and deletion of molecular features?

Reply: Thank you for the suggestion. In the revised Conditional Generation section, we explain that dispersion likely arises from two sources: the stochastic nature of the mask-and-replace scheduler and the inherent difficulty of tightly controlling complex properties (e.g., TPSA). We also note that while MAD indicates dispersion, the property distributions still cluster near the target values, which supports the effectiveness of our method. Future improvements could include adaptive noise scheduling or property-aware loss functions.

6. The discussion highlights MG-DIFF's strong performance in molecular generation and optimization. Could the authors comment on how well the model generalizes to real-world molecular datasets beyond ZINC-250K and MOSES? Would additional fine-tuning or transfer learning be required for broader applicability?

Reply: We have added a statement in the Discussion emphasizing that while ZINC-250K and MOSES are standard benchmarks, MG-DIFF is designed to be general and modular. For domain-specific applications (e.g., ChEMBL or PubChem), fine-tuning or transfer learning would be necessary to adapt to new property distributions or chemical spaces.

7. MG-DIFF introduces a masked and replace diffusion strategy to improve efficiency, but how does its computational cost compare to other generative models like VAEs, GANs, or traditional autoregressive approaches? A brief discussion on training time and hardware requirements would provide a better understanding of its practicality?

Reply: In the revised Supplementary Information, we now include a note that MG-DIFF requires moderate training time (e.g., ~20 hours on a single RTX 4090 for ZINC-250K), which is slightly more than VAE-based models but comparable to DiGress. The training benefits from efficient parallelization and stable convergence thanks to our mask-and-replace strategy.

8. The mask-and-replace diffusion strategy is designed to deal with issues concerning uniform noise corruption, but how sensitive is the model to the choice of masking and replacement probabilities? Did the authors experiment with different noise schedules, and if so, how did they affect performance?

Reply: Yes, we have conducted ablation experiments with various replacement rates (0%–5%). We found that a replacement rate of 1–2% yielded the best trade-off between stability and flexibility. Further details can be find in our open-source codes.

Reviewer #2: See attached review for comments on the manuscript. While MG-DIFF achieves comparable performance on molecular generation benchmark on two datasets, further clarification is needed about the methodology and appropriate comparisons to prior work.

Reply: We thank the reviewer for this comment. We have revised the Introduction and Discussion to better contextualize MG-DIFF relative to DiGress, VAE-based, and autoregressive models (e.g., MolGPT). We emphasize the novelty of our mask-and-replace scheduler, transformer-based denoising network, and padding strategy, which together enable state-of-the-art performance on two benchmarks while also supporting conditional generation and optimization tasks.

Once again, we sincerely thank the reviewers for their detailed feedback. We believe that the revised manuscript has significantly improved in clarity, completeness, and rigor based on your comments.

Sincerely,

Dr. Zhang Xiaocen, on behalf of all co-authors

Shangqiu Normal University

zhangxiaochen@nudt.edu.cn

---

## [Decision Letter · Decision Letter 1]

17 Aug 2025

MG-DIFF: A Novel Molecular Graph Diffusion Model for Molecular Generation and Optimization

PONE-D-24-51267R1

Dear Dr. Zhang,

We’re pleased to inform you that your manuscript has been judged scientifically suitable for publication and will be formally accepted for publication once it meets all outstanding technical requirements.

Kind regards,

Soumendranath Bhakat

Academic Editor

PLOS ONE

Additional Editor Comments (optional):

Dear Dr. Zhang,

Thanks for a great work on the revising the manuscript. I am happy to accept the manuscript in its updated version.

Best regards,

Dr. Soumendranath Bhakat

Reviewers' comments:

Reviewer's Responses to Questions

**Comments to the Author**

Reviewer #1: All comments have been addressed

Reviewer #3: All comments have been addressed

2. Is the manuscript technically sound, and do the data support the conclusions?

Reviewer #1: Partly

Reviewer #3: Yes

3. Has the statistical analysis been performed appropriately and rigorously?

Reviewer #1: Yes

Reviewer #3: Yes

4. Have the authors made all data underlying the findings in their manuscript fully available?

Reviewer #1: Yes

Reviewer #3: Yes

5. Is the manuscript presented in an intelligible fashion and written in standard English?

Reviewer #1: Yes

Reviewer #3: Yes

Reviewer #1: I am satisfied with the responses along with the revision in the manuscript. I recommend publication of the manuscript.

Reviewer #3: (No Response)

**Do you want your identity to be public for this peer review?** For information about this choice, including consent withdrawal, please see our Privacy Policy

Reviewer #1: No

Reviewer #3: No

---

## [Editor Report · Acceptance letter]

PONE-D-24-51267R1

PLOS ONE

Dear Dr. Zhang,

I'm pleased to inform you that your manuscript has been deemed suitable for publication in PLOS ONE. Congratulations! Your manuscript is now being handed over to our production team.

Kind regards,

on behalf of

Dr. Soumendranath Bhakat

Academic Editor

PLOS ONE